# Analysis of R848 as an Adjuvant to Improve Inactivated Influenza Vaccine Immunogenicity in Elderly Nonhuman Primates

**DOI:** 10.3390/vaccines10040494

**Published:** 2022-03-23

**Authors:** Kali F. Crofts, Beth C. Holbrook, Ralph B. D’Agostino, Martha A. Alexander-Miller

**Affiliations:** 1Department of Microbiology and Immunology, Wake Forest School of Medicine, Winston-Salem, NC 27101, USA; kcrofts@wakehealth.edu (K.F.C.); bcholbro@wakehealth.edu (B.C.H.); 2Department of Biostatistical Sciences, Wake Forest School of Medicine, Winston-Salem, NC 27101, USA; rdagosti@wakehealth.edu

**Keywords:** influenza, vaccines, elderly, NHP, adjuvant, R848, antibody

## Abstract

Elderly individuals are highly susceptible to developing severe outcomes as a result of influenza A virus (IAV) infection. This can be attributed to alterations that span the aged immune system, which also result in reduced responsiveness to the seasonal inactivated vaccine. Given the rapidly increasing number of individuals in this age group, it is imperative that we develop strategies that can better protect this population from IAV-associated disease. Based on our previous findings that the TLR7/8 agonist resiquimod (R848) could efficiently boost responses in the newborn, another population with decreased vaccine responsiveness, we evaluated this adjuvant in an elderly African green monkey (AGM) model. AGM aged 16–24 years old (equivalent to 64–96 in human years) were primed and boosted with inactivated A/PuertoRico/8/1934 (H1N1) (IPR8) alone or directly linked to R848 (IPR8-R848). We observed increases in the level of circulating virus-specific IgM antibody 10 days following primary vaccination in AGM that were vaccinated with IPR8-R848, but not IPR8 alone. In addition, there were significant increases in virus-specific IgG after boosting selectively in the IPR8-R848 vaccinated animals. These findings provide insights into the ability of R848 to modulate the aged immune system in the context of IAV vaccination.

## 1. Introduction

Influenza A virus (IAV) infections place a significant burden on public health in the US, with the CDC estimating 12,000–52,000 deaths occurring annually between 2010 and 2020 as a result of IAV infections (https://www.cdc.gov/flu/about/burden/index.html, accessed on 5 January 2022). The elderly have an increased risk of developing severe disease, with individuals aged 65 years and older accounting for 50–70% of seasonal influenza-related hospitalizations and 70–85% of influenza-related deaths every year (https://www.cdc.gov/flu/highrisk/65over.htm, accessed on 5 January 2022). Thus, there is a significant need for strategies that can improve protection from severe disease and death in this vulnerable population.

While influenza vaccination has significantly reduced disease burden in the population as a whole, the immunogenicity of the standard inactivated vaccine is reduced in elderly individuals compared to young adults [1,2], with a review of 67 studies led by Rivetti et al. concluding that standard vaccine effectiveness against influenza-like disease is only 23% in individuals ≥ 65 years of age [3]. There are currently 2 vaccine options approved in the US for people over the age of 65: a high-dose inactivated influenza vaccine or an inactivated influenza vaccine adjuvanted with MF59. The high-dose influenza vaccine has four times the hemagglutinin (HA) antigen present in the standard vaccine. Whilst there are studies supporting higher levels of protection from disease and mortality as a result of high-dose vaccination [4,5], there are also conflicting reports [6]. A study supporting the benefit of the high-dose vaccination showed a 22% reduction in influenza-related visits compared to the standard dose of a vaccine in elderly patients [4]. Another study in individuals over the age of 65 showed significant increases in IAV-specific antibody production in response to high-dose vaccination compared to standard-dose influenza vaccination [5]. An MF59-adjuvanted influenza vaccine has been approved for individuals over the age of 65 in the US based on studies showing improved efficacy in the elderly when compared to standard vaccination [7,8]. The MF59-adjuvanted trivalent inactivated vaccine showed 58% efficacy compared to the non-adjuvanted influenza vaccine, which was ineffective [7]. Although both high-dose and MF59-adjuvanted influenza vaccines have improved efficacy in the elderly compared to standard influenza vaccines, protection rates still remain lower than desired, with a large number of individuals remaining susceptible to severe IAV infection.

Reduced vaccine responsiveness and increased IAV susceptibility in the elderly are both the result of changes that occur in the aging immune system. A well-described characteristic of aging is chronic activation of the immune system that results in higher baseline inflammation, a process known as “inflammaging” [9,10]. Cells of the innate immune system are responsible for the increased inflammation, as they produce proinflammatory cytokines in response to factors associated with aging, i.e., cellular senescence, mitochondrial dysfunction, autophagy, DNA damage, chronic infections such as cytomegalovirus (CMV), or activation of the inflammasome [9]. Cells from elderly individuals also have impairments in dendritic cell (DC) function, including decreased phagocytosis, migration, and cytokine production [11,12,13]. Impaired cytokine production reported in aged DC has been, in part, associated with poor influenza vaccine responsiveness, as seroconversion in the elderly is highly correlated with the ability of DC to produce the appropriate cytokines [12]. The pool of naïve T and B cells is also reduced, with memory cells predominating. Memory T cells often exhibit characteristics of immunosenescence, including decreased proliferation and reduced survival and T cell receptor (TCR) breadth [14]. Studies also support the expansion of T regulatory cells (Tregs) [15], which has been reported to contribute to decreased IAV vaccine responsiveness [16]. Memory B cells are also impacted, exhibiting a reduced ability to differentiate into plasma cells [17]. Together, the impairments that span the innate and adaptive arms of the aging immune system lead to a marked decrease in the production of high-affinity antibodies in response to influenza vaccination [18,19].

As our understanding of accessory signals that drive potent immune responses increases, so has our ability to modulate the immune system to promote effective vaccine responses. One such approach is through the use of toll-like receptor (TLR) agonists as vaccine adjuvants [20]. TLRs are broadly distributed on innate and adaptive immune cells, and their engagement promotes robust cellular activation/maturation following microbial infection. Thus, multiple cell types, including T cells, B cells, and DC, can be modulated by TLR ligands. The majority of TLRs are expressed at the cell surface, but a subset (TLR3, TLR7, TLR8, and TLR9) resides within endosomes [21]. This subset of TLR recognizes nucleic acid structures, i.e., dsRNA (TLR3), ssRNA (TLR7 and 8), and unmethylated CpG (TLR9).

Using a nonhuman primate (NHP) model, our lab has shown that the TLR7/8 agonist resiquimod (R848) directly conjugated to inactivated IAV can induce potent immune responses in newborns [22,23,24,25], mitigating the suboptimal responsiveness to vaccination in this age group. NHPs are a highly relevant animal model for the assessment of this approach as they have a similar TLR distribution and function to humans [26]. Our promising findings in the newborn led us to explore the possibility that this may be a useful approach in the elderly. There is evidence to support the utility of TLR7/8 agonists delivered topically in the context of elderly influenza vaccination. In a randomized, controlled, double-blinded study, individuals between 66.8 and 78.3 years of age received an intradermal inactivated influenza vaccine together with a topical cream that contained imiquimod, an R848-related imidazoquinoline, or a vehicle control. A total of 90% of patients who received the imiquimod seroconverted compared to 39% who received the vehicle control [27].

Here, we investigated the effect of R848 to increase antibodies generated to an inactivated IAV vaccine in an elderly African green monkey (AGM) NHP model.

## 2. Materials and Methods

### 2.1. Animals

AGM aged 16–24 years old (equivalent to a 64–96-year-old human) were used in these studies. Animals were housed at the Vervet Research Colony at the Wake Forest School of Medicine. A total of 7 animals were assigned to each vaccine group based on pre-vaccine influenza-specific antibody (measured within 45 days prior to vaccination) and sex. Assignment ensured similar starting ranges of influenza-specific IgG and similar distributions of male and female animals. The animal care and use protocol was adherent to the US Animal Welfare Act and Regulations and approved by the Institutional Animal Care and Use Committee. The AGM were housed and cared for in accordance with state, federal, and institute policies in facilities accredited by the American Association for Accreditation of Laboratory Animal Care (AAALAC) under standards established in the Animal Welfare Act and the Guide for the Care and Use of Laboratory Animals.

### 2.2. Vaccinations

Animals were vaccinated with inactivated A/PuertoRico/8/1934 (H1N1) (IPR8) (obtained from Charles River) either directly conjugated to R848 (IPR8-R848) [28] or left unconjugated. The R848-conjugated vaccine was prepared by linking a derivative of R848 that was modified to contain a primary amine [24] directly to PR8 virions in a 2-step process using a sulfhydryl cross-linker, GMBS (Thermo Scientific, Waltham, MA, USA), as previously described [28]. R848 was incubated with the GMBS cross-linker for 24 h. The free amine in R848 binds to the N-hydroxysuccinimide portion of the GMBS cross-linker. The R848-GMBS conjugate was then incubated with PR8 for 2 h at 37 °C. The maleimide portion on GMBS binds free thiols on the PR8 virion. PR8-R848 was dialyzed for 2 h and then again overnight before being inactivated with 0.74% formalin. Each vaccine contained 45 μg of PR8. The vaccine was administered intramuscularly in the deltoid muscle of the animal in 500 μL of phosphate-buffered saline (PBS). Animals were boosted 21 days post-vaccination (p.v). Seven animals received IPR8, and seven animals received IPR8-R848.

### 2.3. ELISA for Detection of IAV-Specific Antibody

Nunc MaxiSorp Elisa plates were coated with 1 μg/well of PR8 in sodium carbonate/bicarbonate coating buffer (pH 9.5) overnight at 4 °C. Plates were blocked with 1X Blocking Buffer (10X Casein Blocking Buffer, Sigma-Aldrich, St. Louis, MO, USA) plus 2% goat serum (Lampire Biologicals, Pipersville, PA, USA) for 1 h and then washed. The wash buffer contained PBS with 0.1% Tween 20. Plasma samples were serially diluted in 1X blocking buffer. Wells that contained no virus served as a negative control. Horseradish peroxidase (HRP)-conjugated antibody specific for monkey IgG (Fitzgerald, Acton, CA, USA) or IgM (LifeSpan Biosciences, Seattle, WA, USA) was used to detect bound antibodies. Plates were developed using 3,3′,5,5′-Tetramethylbenzidine dihydrochloride (TMB)(Sigma-Aldrich, St. Louis, MO, USA) and read at 450 nm on a Elx800 Absorbance Microplate Reader (BioTek, Winooski, VT, USA). For each dilution, the OD from the non-virus-coated wells was subtracted from the virus-coated wells. The threshold titer was defined as the value that reached 3X the assay background, i.e., wells that only received 1X blocking buffer + virus (no sample). Fold change was determined by dividing the threshold titer at each given time point by the d0 threshold titer.

### 2.4. ELISA for the Detection of HA Stem-Specific Antibody

ELISA microplates (96-half well) (Greiner bio-one) were coated with 100 ng/well of recombinant A/California/04/2009 (Ca09)(H1N1) stabilized stem [29] in PBS overnight at 4 °C. The plates were blocked with 1X blocking buffer (10X Casein Blocking Buffer, Sigma-Aldrich) plus 2% goat serum (Lampire Biologicals) for 1 h and then washed. Wash buffer contained PBS with 0.1% Tween 20. Plasma samples were serially diluted in 1X blocking buffer. Wells that contained no virus served as a negative control. HRP-conjugated antibodies specific for monkey IgG (Fitzgerald) were used to detect bound antibodies. Plates were developed using TMB (Sigma-Aldrich) and read at 450 nm on a BioTek Elx800 Absorbance Microplate Reader. For each dilution, the OD from the non-virus-coated wells was subtracted from the virus-coated wells. Threshold titer was defined as the value that reached 3X the assay background, i.e., wells that only received 1X blocking buffer + stabilized stem (no sample).

### 2.5. Hemagglutinin Inhibition Assay

AGM plasma was treated with receptor-destroying enzyme (RDE) (Sigma-Aldrich) of Vibrio cholera filtrate overnight at 37 °C. Samples were then heat-inactivated with 2.5% sodium citrate for 30 min at 37 °C. The RDE-treated plasma was then serially diluted and was incubated with 8 HAU/well of PR8 for 30 min before transferring to chicken red blood cells (cRBC) (Lampire Biologicals) for 45 min on ice. Neutralization titers were determined by the last column of wells, where cRBCs formed a button.

### 2.6. Avidity

Avidity assays were performed as the ELISAs described above with the addition of a NaSCN dissociation step following sample incubation. To normalize the total amount of antibody in the assay, the plasma dilution used was determined for each animal based on the dilution that yielded 50% of the MAX OD450 in the ELISA binding curve. Following incubation with plasma, 2-fold dilutions of NaSCN starting at 5M were added to each plate for 15 min. Plates were then washed, and the HRP-conjugated antibody specific for monkey IgG (Fitzgerald) was used to detect bound antibodies and developed like the ELISA assay. The 50% maximum inhibitory concentration (IC_50_) was calculated using GraphPad Prism software.

### 2.7. Statistical Analysis

We conducted power calculations using PASS 13 software where 7 animals in each vaccine group would be required to reach 80% power to detect a difference of 1.86 standard deviations (SD) between the groups using a compound symmetry covariance structure and a correlation between repeated observations of 0.5 with alpha = 0.05 (2-sided). Based on our data, the estimated SD was, at most, 1.5, and the correlation between repeated observations was 0.58. Using these inputs, we can detect a difference of 1.86 titer units with 80% power and alpha = 0.05 (2-sided test). IgM and IgG levels (fold-changes) were compared between (IPR8 vs. IPR8-R848) and within (d0 vs. d10 p.v., d21 p.v., d10 p.b., and d21 p.b.) groups using longitudinal mixed models. In these models, the individual animals were considered as random effects, while time, group, and time-by-group interactions were considered as fixed effects. In order to account for multiple comparisons (with d0 values), Tukey’s method was used to identify significant differences [30]. The mixed-effects models were performed using Proc Mixed in SAS v9.4.

## 3. Results

### 3.1. Elderly Vaccination

AGM aged 16–24 years old (equivalent to 64–96 years in humans) were vaccinated with formalin-inactivated PR8 virus (H1N1) (IPR8) alone or directly linked to a TLR7/8 agonist R848 (IPR8-R848). We utilized an amine derivative of R848 [24] that allowed linkage to an N-hydroxysuccinimide (NHS) group on the GMBS cross-linker. Subsequently, the R848-GMBS was conjugated to the virus via free thiol groups accessible on the surface of the PR8 virion [28]. Following R848 conjugation, the vaccine was inactivated with formalin [28].

We assigned seven animals to each vaccine group based on pre-existing antibody titers and sex to ensure similar starting ranges of influenza-specific IgG and similar distributions of male and female animals (Table 1). Animals were vaccinated with either IPR8 or IPR8-R848 on d0 and were boosted on d21 post-vaccination (p.v.). Femoral blood draws were performed at d10 and 21 p.v. and post-boost (p.b.) to assess circulating levels of IAV-specific antibody and IAV-specific T cell responses. The overall experiment design is shown in Figure 1A.

### 3.2. R848 Adjuvantation Increases IgM Production after Primary Vaccination with Inactivated IAV Vaccine

Firstly, we wanted to assess the kinetics of circulating levels of total IAV-specific IgM in vaccinated AGM over time. IAV-specific IgM was assessed by ELISA using the PR8 H1N1 virus (Figure 1B). We used fold change in antibody as our measure given the differences among animals in the antibody level prior to vaccination. When we compared the two groups over time, we found a significant time by group interaction (*p* = 0.039), suggesting that the change in IgM values over time were different between the 2 groups (IPR8 vs. IPR8-R848). We then examined within-group changes in IgM over time. Here we found that animals vaccinated with IPR8 did not show significant increases in IAV-specific IgM until d10 p.b., and this was lost by d21 p.b. (Figure 1B (left panel)). In contrast, we saw a significant increase in IAV-specific IgM at d10 p.v. in animals that were vaccinated with IPR8-R848 (Figure 1B (right panel)). Thus, the IPR8-R848 group demonstrated a considerably earlier (d10 p.v.) increase in IgM compared to the IPR8 group; however, the slope of the decline in IgM after d10 p.v. was steeper in the IPR8-R848 group than the IPR8 group.

We next measured total IAV-specific IgG in vaccinated AGM over time (Figure 1C). When we compared the two groups over time, there was not a significant difference between the group interaction (*p* = 0.82), suggesting that the change in IgG values over time were not different between the 2 groups (IPR8 vs. IPR8-R848). We then examined within-group changes in IgG over time. A significant increase in IAV-specific IgG fold-change was observed at both d10 and 21 p.b. in IPR8-R848 vaccinated animals (Figure 1C (right panel)). No significant increases were detected in animals vaccinated with IPR8 alone (Figure 1C (left panel)). These data show that IPR8-R848 drives an increase in IgG after the boost. We also evaluated animals at d99-100 following boost. Both groups of animals exhibited a substantial reduction in the fold change from the peak. However, while 3 of 7 animals receiving the IPR8 vaccine had titers higher than at d0, 6 of 7 animals vaccinated with IPR8-R848 had higher antibodies at this late time point.

### 3.3. Elderly AGM Generate HA-Stem Specific IgG in Response to Vaccination

The constant drift in circulating strains and the potential for shift resulting in pandemics have led to a focus on eliciting antibodies to the relatively conserved stem region of HA. These antibodies can provide broader protection than those directed to the HA head [31,32]. In our studies of newborn AGM vaccination, we found that conjugation with R848 resulted in a preferential boost in stem-specific antibodies ([25] and manuscript submitted). To determine whether this occurred in elderly AGM, we measured stem-specific IgG antibodies using a stem-only construct [29]. Figure 2B shows the level of stem-specific antibodies before and after vaccination (d10 p.b.). As with the total IAV-specific response (Figure 2A), nearly all animals showed an increase in stem-specific antibodies, as indicated by the position to the left of the diagonal. Two animals, one in each group, did not show an increase. Perhaps not surprisingly, these were the two animals with the highest pre-vaccination levels. The animals that did have an increase had a similar increase across the group. This is in contrast to the total IAV-specific response, where increases differed appreciably among the animals. Thus, stem-specific antibodies were not increased to a greater extent than non-stem antibodies when R848 was present in the vaccine.

### 3.4. R848-Mediated Increases in IgG Are Independent of Changes in HAI Titer or Avidity

The ability of an antibody to neutralize IAV has long been used as an indicator of protective capacity. We employed the standardly used hemagglutinin inhibition (HAI) assay (Figure 3A) as a measure of neutralizing antibody. Surprisingly, no significant differences were observed after the prime/boost in animals that received IPR8-R848 or IPR8 alone, i.e., changes in HAI titers were not significantly different to baseline levels. It is widely accepted that seroprotection is generated at HAI titers > 40 after influenza vaccination. Interestingly, all animals in both groups had baseline HAI titers starting at levels higher than 1:40. We next measured the avidity of the PR8-specific IgG antibody at d21 p.b. as an indicator of antibody quality (Figure 3B). No difference in IgG avidity was observed between the vaccine groups. Thus, adjuvanting inactivated IAV vaccine with R848 did not improve the neutralizing capacity or the avidity of PR8-specific IgG after prime/boost in elderly AGM.

## 4. Discussion

In this study, we evaluated the potential for R848 to serve as an effective adjuvant for a whole inactivated IAV vaccine in elderly NHP. Elderly individuals are poor responders to the current seasonal influenza vaccines. While high-dose and MF59-adjuvanted vaccines can improve the response, protection does not reach desired levels in this age group. Here we evaluated the ability of an R848-conjugated inactivated IAV vaccine to improve responses in an elderly NHP model. Our rationale for testing this approach was the promising results we observed in newborn NHP [22,23,24,25], another population with an altered immune system that makes eliciting protective responses challenging.

Elderly NHPs represent a valuable model for interrogating age-associated changes in the immune system [33]. Elderly animals in our study had some level of pre-existing antibody to influenza virus, although these levels are low compared to pre-existing immunity in humans. Animals were assigned to groups to similarly distribute those with higher and lower initial levels. The results from our analyses support the ability of R848 to drive earlier increases in IgM and increased levels of IgG antibody following boost. The improved early IgM coupled with the increased IgG at d10 p.b. in animals vaccinated with the R848-containing vaccine supports the ability of this adjuvant to promote antibody responses.

The increase in IgM production in animals that received the IPR8-R848 vaccine at d10 p.v. compared to animals that received the non-adjuvanted vaccine suggests R848 may be promoting early activation of IAV-specific B cell clones. This may occur through direct activation of IAV-specific B cells as a result of TLR7 engagement [34,35]. We have previously shown that an R848-adjuvanted IAV vaccine can promote early B cell activation (24 h following vaccination) in the draining lymph nodes of newborn NHPs [22]. Furthermore, TLR7 engagement on B cells has been reported to support B cell proliferation following IAV infection of mice [36] and can enhance differentiation into antibody-secreting cells in human PBMC [37]. During the early stages of the humoral response, higher affinity B cell clones are known to adopt an extrafollicular plasmablast fate [38]. R848 may facilitate differentiation of B cells with BCR that, in the absence of the signals provided by R848, are not of sufficient affinity to differentiate into antibody-secreting cells. The early increases in IgM may be associated with an augmented primary response that allows for higher IgG levels following boost. R848 may increase the recruitment/activation/proliferation of clones recruited into the response that subsequently undergo isotype switching and differentiation into IgG-secreting cells following secondary antigen exposure. Other potential avenues through which R848 may function are the upregulation of costimulatory molecules on the surface of APC and promotion of proinflammatory cytokine production, resulting in improved cellular and humoral responses [24,28,39,40,41,42].

Interestingly, although we saw increases in IgG in R848-adjuvanted animals after the boost, antibody function (neutralization and avidity) was not improved compared to IPR8-vaccinated animals. This suggests R848 may not be able to overcome the poor GC reactions known to be associated with the aged immune system [19]. Studies have shown that elderly individuals have defects in somatic hypermutation (SHM) as well as class switch-recombination, which contribute to the poorer antibody response elicited by IAV vaccination in this age group [19]. These defects have been attributed to reduced expression of activation-induced cytidine deaminase (AID) and E47 (the key transcription factor that regulates AID) in both mice [43,44] and humans [45]. AID expression has also been correlated to affinity maturation, a process that is essential in the production of high-avidity antibodies. A study of young and elderly adults showed that the level of AID after vaccination correlated with the fold increase in antibody affinity to HA [46]. Affinity maturation to the HA1 region was only observed in the young individuals and was not observed in the elderly.

In our study, we explored the antigen-specific T cell response to vaccination in elderly NHP through stimulation of PBMC (d10 p.b.) with pooled peptides from HA and NA derived from PR8. Both ELISPOT and activation-induced marker (AIM) expression evaluated by flow cytometry were employed. We observed only a minority of animals in each group had detectable T cell responses (data not shown). Of those that did, the magnitude of the response was low and did not differ between the groups. Limitations of this analysis are that a single timepoint was evaluated, and it is restricted to circulating cells. Therefore, further analysis is needed to fully determine the impact of R848 adjuvantation on T cell responses in elderly NHP.

Together, our data support the ability of R848 to increase IgG antibody responses in elderly NHP following vaccination with inactivated IAV. However, it is important to bear in mind that while R848 increases the response, it was modest in comparison to what we observed in newborn NHP [22,23,24,25]. This highlights the critical importance of evaluating potential vaccines in multiple age groups, especially those of the extremes of lifespan. It is possible that a combination adjuvant approach, e.g., R848 administered with another immunostimulatory molecule, may be useful. The Shingrix vaccine, which is highly effective in the elderly, is adjuvanted with AS01_B_, which consists of the saponin QS21 and the toll-like receptor type 4 agonist, MPL. Combining R848 with a saponin or MF59 adjuvant, the latter of which is approved for elderly individuals, may enhance the beneficial effects of R848 and should be explored in future studies.

## Figures and Tables

**Figure 1 vaccines-10-00494-f001:**
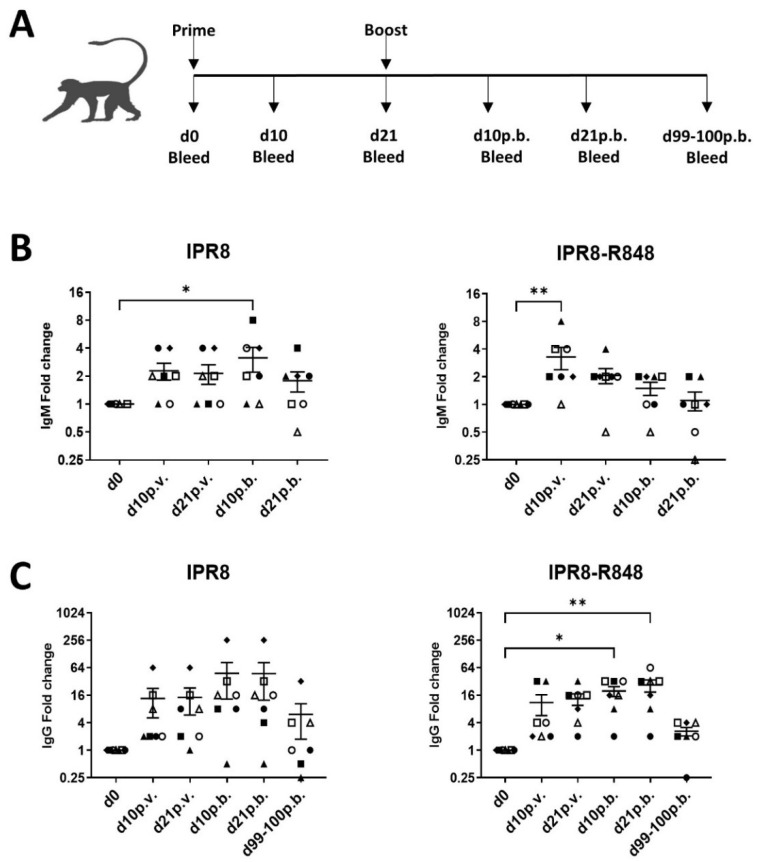
R848 adjuvantation promotes increased levels of antibody after primary vaccination in elderly AGM. (**A**) Elderly AGM were vaccinated with 45 μg of IPR8 (*n* = 7) or R848-adjuvanted IPR8 (*n* = 7) by intramuscular injection. On d21 p.v., animals were boosted with the same vaccine they received as a prime. Blood was drawn on the day of vaccination (d0), d10 p.v., d21 p.v., d10 p.b., d21 p.b., and d99-100 p.b. The levels of circulating PR8-specific IgM (**B**) and IgG (**C**) were quantified from the plasma by ELISA (PR8 coated on well at 1 µg per well). Threshold titer (TT) is defined as the highest dilution at which the sample’s optical density was at least 3 times that of the assay background. Fold change was determined by dividing the threshold titer at each time point by the d0 TT. Each animal is designated by a distinct symbol; these symbols are consistent across time points to allow for longitudinal visualization of each individual response. Antibody levels (fold changes) were compared within (d0 vs. d10 p.v., d21 p.v., d10 p.b., or d21 p.b.) groups using longitudinal mixed models. In these models, the individual animals were considered as random effects, while time, group, and time-by-group interactions were considered as fixed effects. In order to account for multiple comparisons (with d0 values), Tukey’s method was used to identify significant differences. The mixed-effects models were performed using Proc Mixed in SAS v9.4. * *p* = < 0.05, ** *p* = < 0.01.

**Figure 2 vaccines-10-00494-f002:**
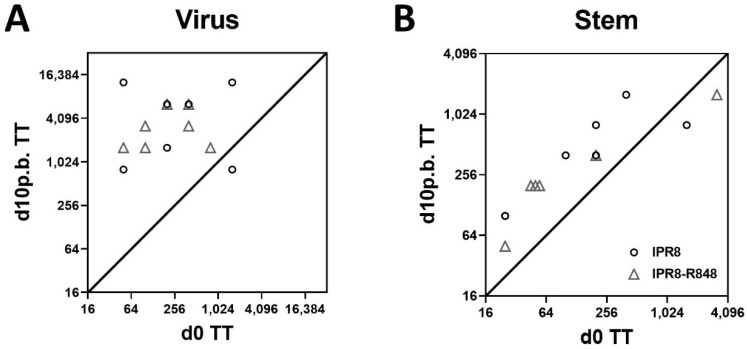
Elderly AGM generates HA-stem-specific IgG in response to vaccination. The levels of circulating IgG were quantified from the plasma of IPR8 (circles)- and IPR8-R848 (triangles)-vaccinated AGM. The d0 threshold titer (TT) for each animal was plotted against the d10 p.b. threshold titer. IgG specific to PR8 (**A**) or Ca09 trimeric stem protein (**B**) was measured. TT is defined by the highest dilution at which the sample’s optical density was at least 3 times that of the assay background.

**Figure 3 vaccines-10-00494-f003:**
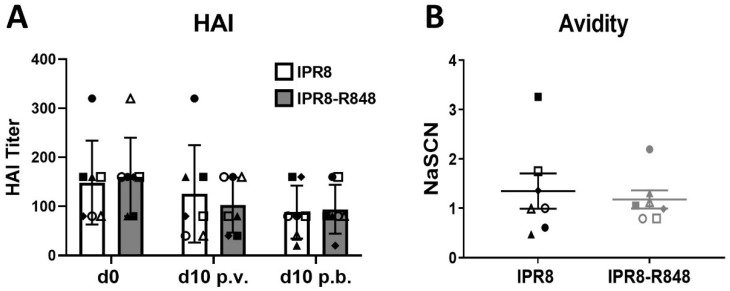
R848 adjuvantation does not improve the neutralizing capacity or avidity of IAV-specific antibodies after prime/boost vaccination in elderly AGM. (**A**) The neutralizing capacity of IAV-specific antibodies was assessed by HAI assays at d0, d10 p.v. and d10 p.b. Serially diluted plasma was incubated with PR8 for 30 min and then transferred to chicken red blood cells for hemagglutination assessment. HAI titer is defined by the highest dilution at which the sample prevents agglutination. Each animal is designated a distinct symbol; these symbols are consistent across time points to allow for longitudinal visualization of each individual response. IPR8- and IPR8-R848-vaccinated animals are designated by black or grey symbols, respectively. (**B**) The average avidity of PR8-specific IgG at d21 p.b. was calculated by determining the NaSCN concentration that gave a 50% reduction in optical absorbance compared to the untreated sample. Seven animals were included in each vaccine group, and the averages shown are the mean ±SEM. Statistical significance was determined using a two-way ANOVA with repeated measures and Tukey’s multiple comparisons test (**A**) or an unpaired two-tailed t-test (**B**). No statistical significance was observed, and the *p*-value was >0.05.

**Table 1 vaccines-10-00494-t001:** Characteristics and threshold titer of vaccine-specific IgG titer prior to vaccination.

Vaccine	Age	Sex	TT IgG
IPR8	18.1	F	1600
	18.9	M	1600
	17	M	200
	19.2	F	50
	18.4	F	400
	17	F	200
	16.2	F	50
IPR8-R848	17.1	F	100
	17.1	F	400
	19.2	M	800
	21.2	F	400
	18.3	M	50
	17.1	F	200
	16.2	F	100

## Data Availability

Not applicable.

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
