# Peer review of "Analysis of R848 as an Adjuvant to Improve Inactivated Influenza Vaccine Immunogenicity in Elderly Nonhuman Primates"

_vaccines, 2022, doi:10.3390/vaccines10040494_

Round 1

Reviewer 1 Report

The manuscript “Analysis of R848 as an adjuvant to improve inactivated influenza vaccine immunogenicity in elderly nonhuman primates” is presenting a new approach of vaccination against Influenza A virus in elderly people. In study, antibody analyses are proving the efficacy of R848 adjuvant vaccine in AGM model. However, study is lacking other characterizations like neutralization assay, HA, and HAI to further reinforce the claims.

Author Response

We are appreciative of the helpful comments from the reviewers, which have improved our manuscript. We have addressed each as described in detail below.

Reviewer 1 comments: “In study, antibody analyses are proving the efficacy of R848 adjuvant vaccine in AGM model. However, study is lacking other characterizations like neutralization assay, HA, and HAI to further reinforce the claims” –We appreciate the importance of functional characterization of antibodies to gain a fuller understanding of their potential roles in vivo. In our study we performed HAI assays (Fig. 3) and assessed avidity (Fig. 3). In both assays, we did not observe statistical changes between the two vaccine groups.

Reviewer 2 Report

The manuscript "Analysis of R848 as an adjuvant to improve inactivated influenza vaccine immunogenicity in elderly nonhuman primates" by Kali F. Crofts et al., Ref: Manuscript Number: vaccines-1628196-peer-review-v1, provides data on the use of TLR7/8 agonist resiquimod (R848) as adjuvant linked to A/PuertoRico/8/1934 (H1N1) (IPR8).  

The authors have described the evaluation of the responses boost of R848 in an elderly in nonhuman primate model, African green monkey (AGM) aged 16-24 years old (equivalent to 64-96 in human years). Overall, 14 animals were divided into two vaccine groups based on the pre-existing antibody titer and sex, and one group received IPR8 alone and the second one IPR8 linked to R848 (IPR8-R848). After the vaccine injection in the deltoid muscle on day 0 and subsequent booster on day 21 post-vaccination, femoral blood draws were performed at day 10 and 21 post-vaccination and post-boost, respectively in the following 10, 21, finally, 99-100 days post boost. In order to evaluate the animal's immune response, different serological tests were carried out, including an ELISA assay for detection of influenza A virus-specific antibodies, an ELISA assay for detection of hemagglutinin (HA) stem-specific antibody, and a hemagglutinin inhibition assay. The materials and methods are straightforward to read and follow. The inclusion of figures and diagrams to complete and clarify the information and the materials and methods are highly appreciated. The results reveal the ability of R848 to modulate the aged immune system in the context of influenza A virus vaccination. Indeed, it was detected a significant increase of IgM at day 10 post-vaccine and IgG after the booster in animals vaccinated with IPR8-R848.

However, the authors do not provide information regarding the presence of local and/or general adverse reactions induced by the injection of the vaccine linked to TLR7/8 agonist resiquimod (R848) used as an adjuvant. This information could be crucial for the further use of the adjuvant in the preparation of inactivated vaccines to be administered to elderly patients.

While this study provides preliminary data that should be further investigated, the general scientific tone of the manuscript sounds good and is worthy to be published after few changes.

Material and methods:

  • Section Animals; lines 98-103: I suggest inserting in this section the overall number of animals involved in the experimentation and the criteria used to divide these into the two different vaccine groups. Moreover, more information concerning the Ethics Committee approbation is requested.
  • Section Statistical analysis; lines 158-164: More information on the power calculation to assess the adequacy of the population size of each investigated group are requested.

Author Response

We are appreciative of the helpful comments from the reviewers, which have improved our manuscript. We have addressed each as described in detail below.

Reviewer 2 comments:

“However, the authors do not provide information regarding the presence of local and/or general adverse reactions induced by the injection of the vaccine linked to TLR7/8 agonist resiquimod (R848) used as an adjuvant.” –We agree with the reviewer that this is an important factor when considering vaccine research for elderly patients. After vaccination, trained veterinary personnel closely evaluated the overall health of the animals for the first 24hrs to monitor any potential side effects from vaccination. No adverse effects were observed after vaccinations. In our previous studies using newborn AGM, only modest increases in CRP were observed at 24h following vaccination and thus we would not anticipate high induction. Unfortunately, we do not have a blood sample from our elderly animals at this time point and thus a more detailed analysis of inflammation could not be performed.   

“Section Animals; lines 98-103: I suggest inserting in this section the overall number of animals involved in the experimentation and the criteria used to divide these into the two different vaccine groups. Moreover, more information concerning the Ethics Committee approbation is requested.” –We appreciate the reviewer’s comments and have revised this section to include more information regarding animal care and housing in the materials and methods lines 101-109. We also included that we adhered to the animal care and use protocol in line 364-365.

“Section Statistical analysis; lines 158-164: More information on the power calculation to assess the adequacy of the population size of each investigated group are requested.” –We apologize that this information was omitted. Our statistician Dr. Ralph D’Agostino performed the power calculations using PASS 13 software and determined that 7 animals in each vaccine group would be required to reach 80% power to detect a difference of 1.86 standard deviations (SD) between the groups using a compound symmetry covariance structure and a correlation between repeated observations of 0.5 with alpha=0.05 (2-sided). Based on our data, the estimated SD was at most 1.5 and the correlation between repeated observations was 0.58. Using these inputs we can detect a difference of 1.86 titer units with 80% power and alpha=0.05 (2-sided test).  

Reviewer 3 Report

The hypothesis is sound, the study well-conducted and the paper well-written. The TLR7/8 agonist resiquimod only had modest effekt the immuneresponse to IAV vaccination. Still it is important to conduct studies to improved the efficacy of influenzavaccines in elderly.

Author Response

We are appreciative of the helpful comments from the reviewers, which have improved our manuscript. 

Reviewer 3 comments: No changes were requested.